# What Type of Patients Did PARAGON-HF Select? Insights from a Real-World Prospective Cohort of Patients with Heart Failure and Preserved Ejection Fraction

**DOI:** 10.3390/jcm9113669

**Published:** 2020-11-15

**Authors:** René Rettl, Theresa-Marie Dachs, Franz Duca, Christina Binder, Fabian Dusik, Benjamin Seirer, Johannes Schönauer, Christina Kronberger, Luciana Camuz Ligios, Christian Hengstenberg, Nina Derkits, Johannes Kastner, Roza Badr Eslam, Diana Bonderman

**Affiliations:** 1Division of Cardiology, Department of Internal Medicine II, Medical University of Vienna, Waehringer Guertel 18-20, 1090 Vienna, Austria; rene.rettl@meduniwien.ac.at (R.R.); theresa-marie.dachs@meduniwien.ac.at (T.-M.D.); franz.duca@meduniwien.ac.at (F.D.); christina.binder@meduniwien.ac.at (C.B.); fabian.dusik@meduniwien.ac.at (F.D.); benjamin.seirer@meduniwien.ac.at (B.S.); n1542278@students.meduniwien.ac.at (J.S.); n1541334@students.meduniwien.ac.at (C.K.); luciana.camuzligios@meduniwien.ac.at (L.C.L.); christian.hengstenberg@meduniwien.ac.at (C.H.); johannes.kastner@meduniwien.ac.at (J.K.); 2Novartis Pharma GmbH, Stella-Klein-Loew-Weg 17, 1020 Vienna, Austria; nina.derkits@novartis.com; 3Division of Cardiology, Klinik Favoriten, Kundratstraße 3, 1100 Vienna, Austria

**Keywords:** heart failure with preserved ejection fraction, HFpEF, PARAGON-HF, real-world, outcomes

## Abstract

The PARAGON-HF clinical trial suggested that sacubitril/valsartan may become a treatment option for particular subgroups of patients with heart failure and preserved ejection fraction (HFpEF). However, the proportion of real-world HFpEF patients who are theoretically superimposable with the PARAGON-HF population is yet unknown. The present study was performed to define the proportion of real-world PARAGON-HF-like patients and to describe their clinical characteristics and long-term prognosis in comparison with those who would not meet PARAGON-HF criteria. We systematically applied PARAGON-HF inclusion and exclusion criteria to a total of 427 HFpEF patients who have been participating in a prospective national registry between December 2010 and December 2019. In total, only 170 (39.8%) registry patients were theoretically eligible for PARAGON-HF. Patients not meeting inclusion criteria (41.0%) were less impaired with respect to exercise capacity (median 6-min walk distance: 385 m (IQR: 300–450) versus 323 m (IQR: 240–383); *p* < 0.001) had lower pulmonary pressures (mean pulmonary artery pressure (mPAP): 31.2 mmHg, standard deviation (SD): ±10.2 versus 32.8 mmHg, SD: ±9.7; *p* < 0.001) and better outcomes (log-rank: *p* < 0.001) as compared to the PARAGON-like cohort. However, patients theoretically excluded from the trial (19.2%) were those with most advanced heart failure symptoms (median 6-min walk test: 252 m (IQR: 165–387); *p* < 0.001), highest pulmonary pressures (mPAP: 38.2 mmHg, SD: ±12.4; *p* < 0.001) and worst outcome (log-rank: *p* = 0.037). We demonstrate here that < 40% of real-world HFpEF patients meet eligibility criteria for PARAGON-HF. We conclude that despite reasons for optimism after PARAGON-HF, a large proportion of HFpEF patients will remain without meaningful treatment options.

## 1. Introduction

The general prevalence of heart failure with preserved ejection fraction (HFpEF), as well as its relative prevalence compared to heart failure with reduced ejection fraction (HFrEF), have increased in recent years. Meanwhile, the condition accounts for more than 50 percent of all heart failure (HF) cases [1,2,3]. The complex pathophysiology of HFpEF is still incompletely understood, and the underlying phenotypic heterogeneity is far greater than in HFrEF [4]. For example, Shah et al. were able to classify HFpEF patients into three subgroups with distinct clinical characteristics and prognosis [5]: (I) the *hypertension phenotype*, which represents younger patients with moderate diastolic dysfunction; (II) the *obesity phenotype*, which includes obese, diabetic patients with a high prevalence of sleep apnea and impaired left ventricular (LV) relaxation; and (III) the *right heart phenotype*, which is composed of older patients with significant chronic kidney disease (CKD) and pulmonary hypertension (PH).

Despite these very important insights into the rather complex and heterogeneous pathophysiology groups, until today, specific medical treatments targeting either the left or the right ventricle (RV) failed to demonstrate any clinical benefit in this patient population [6]. However, the recently published multicenter, randomized, double-blind, parallel-group, active-controlled, phase III study PARAGON-HF gave legitimate reason for optimism with regards to potential new therapeutic armamentarium for affected patients. In brief, the trial evaluated the efficacy and safety of sacubitril/valsartan compared to valsartan, on morbidity and mortality in 4822 HF patients (HFpEF and heart failure with mid-range ejection fraction (HFmrEF)) [7]. Although the primary composite endpoint defined as hospitalization for HF or cardiovascular death was narrowly missed, two of the twelve pre-specified subgroups showed possible heterogeneity of treatment effect, with a suggestion of benefit. This was seen in particular in patients with an ejection fraction (EF) ≤ 57% and in women, who represent a high proportion of patients with HFpEF [8,9], making sacubitril/valsartan a promising drug.

In an attempt to enrich the study cohort with patients who had a high likelihood of reaching at least one of the co-primary endpoints, inclusion criteria allowed only patients with more advanced disease as mirrored by a requirement of diuretic treatment and relatively high serum levels of N-terminal prohormone of brain natriuretic peptide (NT-proBNP), if recent hospitalization for HF was absent. On the other hand, a series of exclusion criteria had to be observed by the international study sites, in order to guarantee patient safety. For example, patients with CKD stage 4 and higher were not able to participate in PARAGON-HF. Furthermore, patients with probable alternative diagnoses that could have accounted for their symptoms, such as pulmonary diseases, anemia, or morbid obesity, as well as patients with a life expectancy <3 years were excluded.

The present study was performed to systematically investigate the proportion of real-world HFpEF patients who are theoretically superimposable with the PARAGON-HF population and to describe their clinical characteristics and long-term prognosis in comparison with those not eligible for the study.

## 2. Material and Methods

### 2.1. Study Design

In the present study, we systematically applied PARAGON-HF inclusion and exclusion criteria to a total of 427 HFpEF patients who have been participating in a prospective national registry between December 2010 and December 2019. The registry complies with the Declaration of Helsinki and was approved by the ethics committee of the Medical University of Vienna (EK #796/2010). All patients gave written informed consent before enrollment.

The diagnosis of HFpEF was made according to current consensus statements of the European Society of Cardiology [10] and American Heart Association [11]. For registry inclusion, the following criteria were used: (I) signs and symptoms of HF, (II) left ventricular ejection fraction (LVEF) ≥ 50%, (III) NT-proBNP > 220 pg/mL, and (IV) evidence of structural heart disease defined by left atrial (LA) enlargement (LA volume index > 34 mL/m^2^) or LV hypertrophy (LV mass index ≥ 115 g/m^2^ for males and ≥95 g/m^2^ for females) or evidence of LV diastolic dysfunction by transthoracic echocardiography (TTE) assessed via the ratio of early transmitral blood velocity (E) to early diastolic mitral annular velocity (e’). LV diastolic dysfunction was diagnosed in patients with an E/e’ ratio > 15. If the diagnosis of HFpEF was likely according to clinical, laboratory and imaging parameters, right heart catheterization (RHC) was performed in order to confirm the diagnosis by a pulmonary artery wedge pressure (PAWP) > 12 mmHg [12]. In addition, the following exclusion criteria were applied: (I) significant coronary artery disease, defined as a visual stenosis > 50% in one of the main vessels and/or over 70% in one of the distal vessels, assessed by coronary angiography; (II) significant valvular or congenital heart disease; (III) end-stage renal disease, defined as estimated glomerular filtration rate (eGFR) < 15 mL/min/1.73 m^2^; and (IV) cardiac amyloidosis (CA), which was diagnosed in accordance with current recommendations, where endomyocardial biopsies were only necessary, if non-invasive test results, including cardiac magnetic resonance (CMR) imaging, TTE, bone scintigraphy, serum immunofixation, urine immunofixation, and serum free light chain assay were ambiguous [13]. For comparison, we have listed most relevant differences with respect to inclusion and exclusion criteria between our registry and PARAGON-HF (Table 1).

### 2.2. Baseline Assessment of the Real-World Cohort

Baseline assessment included clinical evaluation, routine blood parameters, exercise capacity, cardiac imaging, and hemodynamic characterization by RHC.

#### 2.2.1. Clinical and Laboratory Assessment

Functional status was defined by New York Heart Association (NYHA) functional class. A 6-min walk test (6-MWT) was performed to assess submaximal exercise capacity in accordance with the American Thoracic Society Guidelines [14]. NT-proBNP serum levels were determined in all registry participants.

#### 2.2.2. Cardiac Imaging

Patients underwent TTE on high-end machines (GE Vivid E95 and Vivid 7, GE Healthcare, Wauwatosa, WI, USA) by board-certified and experienced operators in accordance with current recommendations [15,16]. Image analysis was performed on a modern offline clinical workstation equipped with dedicated software (EchoPAC, GE Healthcare, Wauwatosa, WI, USA).

CMR imaging was performed in patients without respective contraindications on a 1.5-T scanner (MAGNETOM Avanto, Siemens Healthcare, Erlangen, Germany), consisting of standard protocols that included functional and late gadolinium enhancement imaging (0.1 mmol/kg gadobutrol (Gadovist, Bayer Vital, Leverkusen, Germany)), if eGFR was ≥ 30 mL/min/1.73 m^2^ [17].

#### 2.2.3. Invasive Hemodynamic Assessment

Invasive hemodynamic parameters were assessed by RHC with a 7-French Swan-Ganz catheter (Edwards Lifesciences, Irvine, CA, USA). Filling pressures were calculated as the average over eight heart cycles. Parameters of interest were systolic pulmonary artery pressure, diastolic pulmonary artery pressure, mean pulmonary artery pressure (mPAP), PAWP, right atrial pressure, cardiac index, and stroke volume index. The diastolic pressure gradient and pulmonary vascular resistance were calculated according to standard formulae [18].

### 2.3. Definition of Clinical Outcomes

Clinical outcomes were either ascertained by follow-up visits at our HFpEF outpatient clinic or phone calls in case of immobility. The composite endpoint parameter was defined as either hospitalization for HF or death from cardiac causes. When an event occurred, local and external medical records were carefully screened and reviewed by a clinical adjudication committee of cardiology specialists (D.B., R.B.E.). Hospitalization for HF was defined as hospital admission due to clinical signs of acute cardiac decompensation. Death from cardiac causes was defined as: (I) presence of RV or LV dysfunction assessed by TTE and/or CMR; (II) presence of clinical signs of decompensated HF at the time of death such as dyspnea, peripheral edema, ascites, hepatic failure, or jugular venous distension; or (III) documented life-threatening arrhythmias, such as ventricular tachycardia or fibrillation.

### 2.4. Statistical Analysis

A systematic analysis of registry data with respect to PARAGON-HF eligibility was carried out manually. Continuous variables are expressed as either means and standard deviations or median and interquartile ranges (IQR). Categorical variables are presented as numbers and percent. Analysis of variance (ANOVA), Kruskal–Wallis test, or chi-square test were used to compare between multiple cohorts, while the t-Test, Mann–Whitney–U test, or chi-square test were used to compare between two cohorts. In all calculations, a *p*-value of < 0.05 was considered to be statistically significant. Kaplan–Meier plots (log-rank test) were used to verify time-dependent discriminative power of parameters of interest. All statistical analyses were performed with SPSS version 26 (IBM Corp, New York, NY, USA).

## 3. Results

### 3.1. Baseline Characteristics

#### 3.1.1. Real-World HFpEF Cohort

On an average, patients in our real-world HFpEF cohort were 72.0 years old (SD: ±8.4); they were predominately female (70.0%) and presented with typical comorbidities, such as arterial hypertension (93.9%), atrial fibrillation (AF) (58.1%), and diabetes mellitus type 2 (34.2%). For better comparison, we have listed the most important clinical characteristics of our cohort together with PARAGON-HF baseline characteristics (Table 2). Noticeable between-cohort differences were encountered in gender distribution (female: 70.0% versus 51.7%). Other differences were directly attributable to distinct eligibility criteria. For example, the proportion of patients in NYHA functional class III in our registry was higher as compared to PARAGON-HF (55.7% versus 19.4%), mean LVEF were higher in our registry patients (60.3%, SD: ±7.9 versus 57.6%, SD: ±7.9), while the rate of prior HF hospitalizations was higher in PARAGON-HF patients (21.3% versus 48.1%). AF occurred more frequently in the registry (58.1% versus 32.4%) because of a trial-immanent capping of AF patients.

When applying PARAGON-HF inclusion and exclusion criteria to our real-world cohort, four subgroups could be identified: (I) those theoretically eligible for PARAGON-HF (*Cohort 1*, *real-world PARAGON*, *n* = 170); (II) those who would have missed inclusion criteria (*Cohort 2*, *n* = 119); (III) those who would have met inclusion criteria but would have fulfilled one or more exclusion criteria (*Cohort 3*, *n* = 82); and finally (IV) an overlap subgroup of patients who missed inclusion and also met exclusion criteria (*n* = 56) that were assigned to *Cohort 2.* A flowchart demonstrating the hypothetical eligibility for PARAGON-HF and a detailed description of patients who missed the inclusion and/or fulfilled the exclusion criteria are shown in Figure 1 and Figure 2, respectively.

#### 3.1.2. Real-World PARAGON-HF-Like Cohort

Detailed baseline characteristics comparing the three cohorts are shown in Table 3 and in Appendix A. Most of the 170 patients who were theoretically eligible for PARAGON-HF were female (68.2%) with an average age of 73.3 years (SD: ±7.4). We detected clear differences regarding clinical characteristics between real-world PARAGON-HF-like patients (*Cohort 1*) and patients who would have missed inclusion (*Cohort 2*), as well as those would have fulfilled exclusion criteria (*Cohort 3*). With respect to the proportion of patients in NYHA functional class ≥ III, real-world PARAGON-HF-like patients (*Cohort 1*) reported more often severe exertional dyspnea than patients in *Cohort 2* but were less breathless as compared to *Cohort 3* (*p* < 0.001). Likewise, patients in *Cohort 1* covered a shorter walking distance in six minutes than patients in *Cohort 2* but showed better exercise capacity as compared to *Cohort 3* (*p* < 0.001, Figure 3). Analysis of cardiac biomarkers yielded higher median serum levels of NT-proBNP in PARAGON-HF-like patients (*Cohort 1*) as compared to *Cohort 2*, while patients in *Cohort 3* showed the highest median serum levels of NT-proBNP (*p* < 0.001).

When looking at the three distinct cohorts from an imaging and hemodynamic angle, significant between-group differences regarding right heart dimensions and function could be detected. In detail, patients who were theoretically eligible for PARAGON-HF (*Cohort 1*) had more advanced RV dysfunction than patients in *Cohort 2* but showed better RV function compared to *Cohort 3* (TAPSE: *p* = 0.001, RVEF: *p* = 0.012; Table 3 and Appendix A). Likewise, real-world PARAGON-HF-like patients (*Cohort 1*) presented with higher pulmonary and intracardiac filling pressures as compared to *Cohort 2.* However, pulmonary hemodynamic impairment was most pronounced in *Cohort 3* (mPAP: *p* < 0.001, PAWP: *p* < 0.001).

### 3.2. Outcome According to Eligibility for PARAGON-HF

During a median follow-up period of 47.0 months (IQR: 24.0–83.0), a total of 135 (31.6%) patients had reached the composite endpoint of either HF hospitalization or death from cardiac causes. Considering the individual components of the composite endpoint, 122 (28.6%) patients were hospitalized for HF, while 13 (3.0%) patients died of cardiac causes within the follow-up interval without prior HF hospitalization.

We detected significant differences with respect to cardiac outcome parameters between patients who were theoretically eligible for PARAGON-HF (*Cohort 1:* composite endpoint n = 65 (38.2%), hospitalization for HF n = 60 (35.3%), death from cardiac causes n = 23 (13.5%)) and those who would have missed inclusion (*Cohort 2:* composite endpoint n = 30 (17.1%), hospitalization for HF n = 26 (14.9%), death from cardiac causes n = 6 (3.4%)) as well as those who would have fulfilled exclusion criteria (*Cohort 3:* composite endpoint n = 40 (48.8%), hospitalization for HF n = 36 (43.9%), death from cardiac causes n = 16 (19.5%)) with *p*-values < 0.001.

Mirroring their clinical in-between position compared to *Cohorts 2 and 3*, real-world PARAGON-HF-like patients faced a dismal prognosis with respect to cardiac outcome parameters against the background of *Cohort 2* (Figure 4, log-rank: *p* < 0.001; Appendix A, log rank: *p* = 0.001; Appendix A, log rank: *p* = 0.001; Appendix A, log-rank: *p* = 0.005) but were superior to patients in *Cohort 3* (Figure 4, log-rank: *p* = 0.037; Appendix A, log rank: *p* = 0.085; Appendix A, log rank: *p* = 0.177, Appendix A, log-rank: *p* = 0.140). However, no differences were observed regarding non-cardiac outcome parameters (Appendix A, log rank: *p* = 0.629 and *p* = 0.832).

## 4. Discussion

The recently published PARAGON-HF trial yielded encouraging results with respect to a potentially beneficial outcome of HFpEF patients treated with sacubitril/valsartan. However, as a PARAGON-HF trial site, our general impression was that only a minority of patients fulfilling the diagnostic criteria of HFpEF were eligible for the study, based on given inclusion and exclusion criteria. The present study was undertaken in order to systematically define the proportion of real-world HFpEF patients that are congruent with the PARAGON-HF cohort, and to define the clinical phenotype and long-term prognosis of the PARAGON-HF-like population as opposed to HFpEF patients not eligible for the PARAGON-HF trial.

Using our data base that has been documenting clinical, imaging, and hemodynamic characteristics of a prospectively followed HFpEF cohort, we were able to demonstrate that: (I) The proportion of HFpEF patients who were theoretically eligible for PARAGON-HF was only 39.8% (corresponding to 170 out 427 patients). (II) Not unexpectedly, a similar proportion of patients (41.0%) were theoretically precluded from the PARAGON-HF trial, mostly due to low NT-proBNP serum levels and/or lack of requirement of diuretic treatment. (III) Only a minority of patients (19.2%) would have been excluded from the trial, mostly due to obesity, anemia, CKD stage 4 and higher, severe hypo- or hypertension, or combined causes.

When looking at the three distinct cohorts from an imaging and hemodynamic angle, significant between-group differences could be detected with respect to right heart dimensions and function (Table 3). Patients who theoretically fulfilled one or more exclusion criteria (*Cohort 3*) were more advanced with respect to RV dysfunction parameters as compared to *Cohorts 1 and 2* (TAPSE: *p* = 0.001, RVEF: *p* = 0.012; Table 3 and Appendix A). Likewise, pulmonary hemodynamic impairment was most pronounced in *Cohort 3*. This translated into highly impaired exercise capacity as measured by the 6-MWT (*p* < 0.001).

Most importantly, this cohort also showed the most dismal prognosis with regard to the composite endpoint, as well as its single components (hospitalization for HF, death from cardiac causes) when compared with *Cohort 1* (Figure 4, log-rank: *p* = 0.037; Appendix A, log-rank: *p* = 0.085; Appendix A, log-rank: *p* = 0.177) and *Cohort 2* (Figure 4, log-rank: *p* < 0.001; Appendix A, log-rank: *p* = 0.001; Appendix A, log-rank: *p* = 0.001).

Interestingly, no between-cohort differences were found with respect to geometry and function of the LV. However, we observed a gradual difference in LV filling pressures, with highest values in *Cohort 3* and lowest in *Cohort 2* (*p* < 0.001, Table 3).

The three PARAGON-HF-defined phenotypes show clear similarities with machine learning-derived phenotypes as described by Shah et al. [5]. From a clinical perspective, the most interesting finding was that the PARAGON-HF-defined *Cohort 3* in the present work showed striking similarities with the so-called phenogroup 3 that was characterized by lowest GFR, PH and most severe RV dysfunction. These patients were precluded from PARAGON-HF. Less sharp separating lines were detected with respect to Shah’s phenogroup 2; patients from both *Cohorts-1 and 2-* seemed to overlap with this metabolic phenotype. Phenogroup 1, according to Shah et al., that included younger patients with relatively normal cardiac biomarkers was underrepresented in our study, given higher NT-proBNP entrance criteria for our registry.

### 4.1. Differences between the Real-World PARAGON-HF-Like Cohort and the PARAGON-HF Cohort

We detected a notable difference in the proportion of NYHA functional class II and III patients between our real-world PARAGON-HF-like cohort and the PARGON-HF study cohort (NYHA II: 31.8 versus 77.2%, NYHA III: 61.7% versus 19.4%). This difference could be explained by our patient inclusion process for our registry, which demanded a PAWP > 12 mmHg. Therefore, we probably selected a higher proportion of patients with significantly elevated left-sided filling pressures, which in turn correlates with the severity of the dyspnea [19].

The most striking difference between our real-world PARAGON-HF-like cohort and the original PARAGON-HF cohort was encountered with respect to gender distribution (female gender: 68.2% versus 51.7%). Female predominance in HFpEF has been reported in former studies [8,20]. One explanation for this discrepancy with respect to PARAGON-HF may be that patients with a diagnosis of CA, which affects predominately male patients [21], were not explicitly excluded from PARAGON-HF, while every single patient in our registry underwent CA work-up before inclusion. In this context, the PARAGON-HF authors also assumed that amyloid cardiomyopathy may have accounted for the reduced responsiveness of sacubitril-valsartan in patients with higher EF [7]. Beyond that, there is a well-known male overrepresentation in cardiovascular trials [22].

Although the PARAGON-HF trial showed no significant beneficial effect of treatment with sacubitril/valsartan in the entire study cohort, the subgroup analysis indicated a potential benefit, particularly in lower EF ranges (≤ 57%) and in women. In light of the fact that both subgroups had a high likelihood of benefitting from treatment in PARAGON-HF, potential consequences for effective therapeutic interventions in the real world are still unclear. Per current definition, HFpEF patients, on average, have higher EFs and therefore may be less responsive to therapy as compared to the PARAGON-HF cohort, which also included HFmrEF patients. On the other hand, there was a female predominance in the real-world cohort as compared to the PARAGON-HF patients, favoring the notion that more patients in the real world than in PARAGON-HF could benefit from treatment with sacubitril/valsartan.

### 4.2. Limitations

We are fully aware of all limitations that are associated with the single-center design of the registry. Compared to PARAGON-HF, our study cohort was relatively small, and the number of events was limited. However, even though, a center-specific bias cannot be excluded, limiting data collection to one center has the advantage of consistency of diagnostic and clinical work-up, as well as follow-up. Due to the systematic use of imaging and hemodynamic assessments, we were able to study a very well-characterized pure HFpEF population. Furthermore, our results are based on a comparison of a single-center HFpEF cohort with a multinational study, and patient populations may vary regionally.

One of the major limitations of our study concerns the inclusion criteria, which follow the current definition for HFpEF, while PARAGON-HF investigators chose to expand the trial to a subpopulation of HFmrEF patients. Another limitation may be due to the fact that the registry did not cover the entire spectrum of HFpEF patients. In particular, the so-called “hypertension phenogroup” according to Shah et al. [5], which also included younger patients with moderate diastolic dysfunction and only mildly increased serum levels of NT-proBNP, may have been precluded from the registry due to rather stringent inclusion criteria. A further limitation of the present study was that we did not differentiate between different types of atrial fibrillation (paroxysmal, persistent, long-persistent, permanent). However, there was no differentiation between the types of atrial fibrillation in the PARAGON-HF trial either.

In terms of clinical endpoints, 38.2% of registry patients in the PARAGON-like cohort but only 22.6% of patients in PARAGON-HF had experienced at least one outcome event. Comparability may be hampered by differences related to follow-up with a median of 47 months in the registry versus 35 months in PARAGON-HF. Furthermore, we did not investigate any admissions for non-cardiac reasons. However, this particular endpoint was not recorded in the PARAGON-HF trial either.

## 5. Conclusions

The PARAGON-HF clinical trial suggested that sacubitril/valsartan may become a treatment option for particular subgroups of patients who have been diagnosed with HFpEF.

We demonstrate here that less than 40% of the broad phenotypic spectrum of HFpEF patients were compatible with the stringent PARAGON-HF inclusion criteria. In fact, despite the previous use of angiotensin converting enzyme inhibitors (ACEi) or angiotensin receptor blockers (ARB), which was the most difficult criterion to be met when applying PARADIGM to real world, it was not necessary to be enrolled in PARAGON-HF; 69.6% of patients in our real-world HFpEF cohort had been treated with ACEi/ARB, but still most of them did not meet the trial criteria. Most importantly, patients with most pronounced pulmonary vascular disease, severe RV dysfunction, and most dismal prognosis were precluded from PARAGON-HF. Whether these patients would benefit from sacubitril/valsartan remains speculative.

In addition to the fact that PARAGON-HF did not demonstrate a clear benefit of the use of sacubitril/valsartan in patients with HFpEF, the present study revealed that a large proportion of real-world HFpEF patients do not meet the trial criteria and therefore would not benefit from this treatment option. However, a study focusing on the real-world applicability of PARAGON-HF in patients with mildly reduced ejection fraction (45–57%) would be advisable, since these are patients which are most likely to benefit from treatment with sacubitril/valsartan.

## Figures and Tables

**Figure 1 jcm-09-03669-f001:**
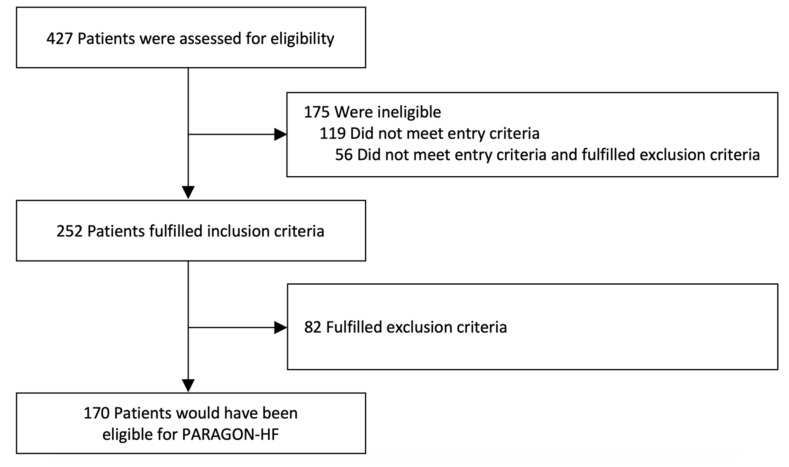
Flowchart showing hypothetical eligibility for PARAGON-HF. Patients (*n* = 427) from a real-world HFpEF registry were tested for their hypothetical eligibility for PARAGON-HF.

**Figure 2 jcm-09-03669-f002:**
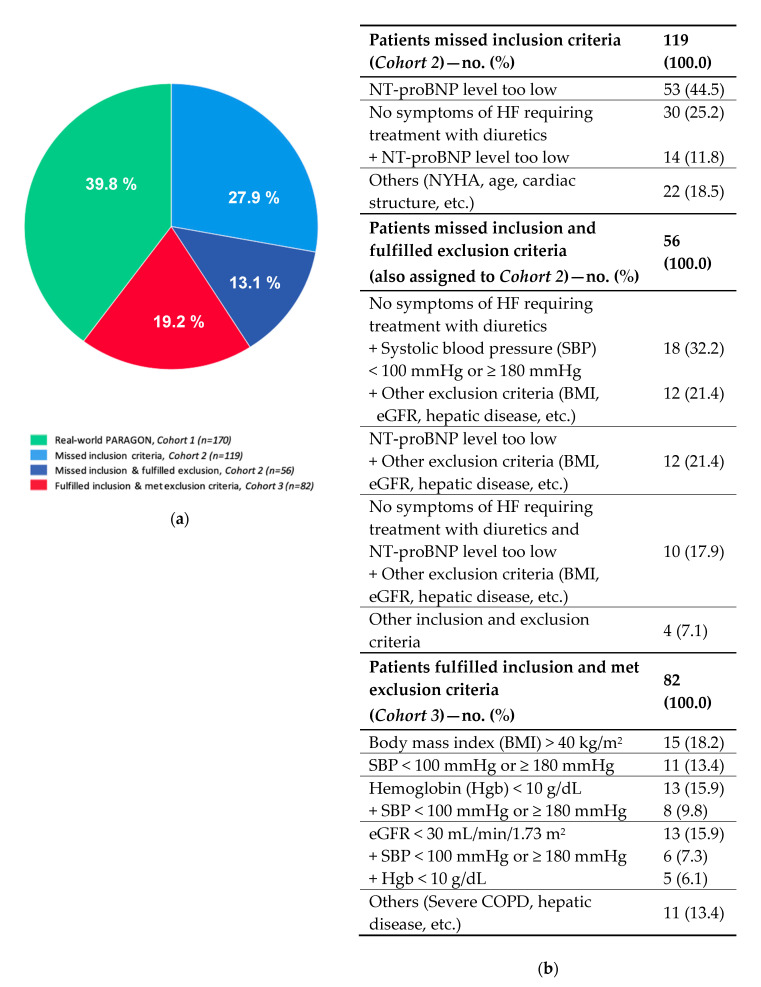
Main reasons for missing inclusion or fulfilling exclusion criteria of PARAGON-HF. (**a**) Represents a graphical description of our real-world HFpEF cohort according to PARAGON-HF eligibility. (**b**) Shows a detailed listing of reasons for missing inclusion and/or fulfilling exclusion criteria of PARAGON-HF. Values are given as total numbers and percent. NT-proBNP indicates N-terminal prohormone of brain natriuretic peptide; HF, heart failure; NYHA, New York Heart Association; BMI, body mass index; eGFR, estimated glomerular filtration rate; COPD, chronic obstructive pulmonary disease.

**Figure 3 jcm-09-03669-f003:**
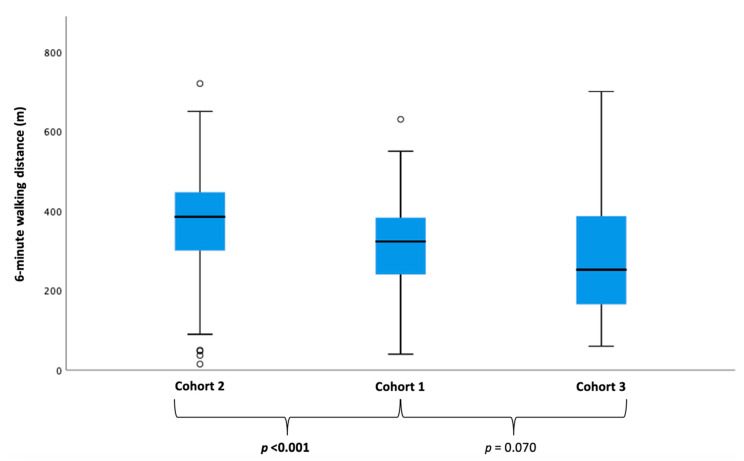
Between-cohort differences in exercise capacity. Boxplots showing median 6-min walking distances of the PARAGON-like proportion of the real-world HFpEF cohort (*Cohort 1*), patients who did not meet the inclusion criteria (± fulfilled exclusion criteria) (*Cohort 2*) and patients who would have been excluded (*Cohort 3*). Kruskal–Wallis test was performed and showed a significant difference across the different cohorts (*p* < 0.001).

**Figure 4 jcm-09-03669-f004:**
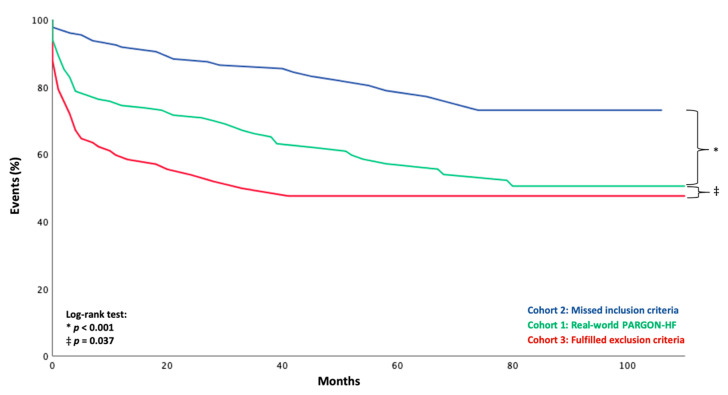
Kaplan–Meier plots for the composite endpoint of hospitalization for heart failure or death from cardiac causes according to eligibility for PARAGON-HF.

**Table 1 jcm-09-03669-t001:** Comparison of most relevant differences in inclusion and exclusion criteria between the real-world heart failure and preserved ejection fraction (HFpEF) registry and PARAGON-HF.

	Real-World HFpEF Registry	PARAGON-HF
**Inclusion criteria**		
Clinical status	Signs and symptoms of HF	Symptoms of HF requiring treatment with diuretics
NT-proBNP	>220 pg/mL	>300 pg/mL or >900 pg/mL in AF, resp. or HF hospitalization and >200 pg/mL or >600 pg/mL in AF, resp.
Echocardiographic findings	LVEF ≥ 50% Evidence of structural heart disease (one of the following): LA enlargement defined by LAVI > 34 mL/m^2^ LVH defined by LVMI ≥115 g/m^2^ for males and ≥95 g/m^2^ for females OR Evidence of LV diastolic dysfunction defined by: E/e’ ratio > 15	LVEF ≥ 45% Evidence of structural heart disease (one of the following): LA enlargement defined by LA width ≥ 3.8 cm, or LA length ≥ 5.0 cm, or LA area ≥ 20 cm^2^, or LAVI ≥ 29 mL/m^2^ LVH defined by IVS ≥ 1.1 cm
Invasive hemodynamic parameters	PAWP > 12 mmHg	Not assessed
**Exclusion criteria**		
Cardiovascular diseases	Significant CAD	ACS or urgent PCI within 3 months prior to Visit 1 MI or CABG within 6 months prior to Visit 1
Cardiac amyloidosis	Exclusion	Not specifically mentioned
Pulmonary diseases	Primary pulmonary HTN Severe pulmonary disease requiring oxygen therapy	Primary pulmonary HTN Severe pulmonary disease incl. COPD (i.e., requiring oxygen therapy or chronic therapies)
Systolic blood pressure	No restrictions	SBP ≥ 180 mmHg SBP > 150 and < 180 mmHg (unless 3 or more aHTN drugs) SBP < 110 mmHg
Anemia	No restrictions	Hemoglobin < 10 g/dL
Obesity	No restrictions	Body mass index > 40 kg/m^2^
Chronic kidney disease	Stage 5 (eGFR < 15 mL/min/1.73 m^2^)	Stage 4 (eGFR < 30 mL/min/1.73 m^2^) and higher

HF indicates heart failure; NT-proBNP, N-terminal prohormone of brain natriuretic peptide; AF, atrial fibrillation; LVEF, left ventricular ejection fraction; LV, left ventricular; E/e’, ratio of early transmitral blood velocity to early diastolic mitral annular velocity; LA, left atrial; LAVI, left atrial volume index; LVH, left ventricular hypertrophy; LVMI, left ventricular mass index; IVS, interventricular septal thickness; PAWP, pulmonary artery wedge pressure; CAD, coronary artery disease; ACS, acute coronary syndrome; PCI, percutaneous coronary intervention; MI, myocardial infarction; CABG, coronary artery bypass graft; HTN, hypertension; COPD, chronic obstructive pulmonary disease; SBP, systolic blood pressure; aHTN, arterial hypertension; eGFR, estimated glomerular filtration rate.

**Table 2 jcm-09-03669-t002:** Baseline characteristics of the real-world HFpEF cohort as compared to PARAGON-HF.

Characteristic	Real-World HFpEF Cohort n = 427	PARAGON-HF Cohort n = 4796
Age—years	72.0 ± 8.4	72.8 ± 8.4
Female sex—no. (%)	299 (70.0)	2479 (51.7)
Systolic blood pressure—mmHg *	140.0 ± 21.3	130.6 ± 15.5
Heart rate—beats/min	71.6 ± 14.1	70.5 ± 12.3
Body-mass index—kg/m^2^	30.0 ± 6.3	30.3 ± 5.0
Serum creatinine—mg/dL	1.2 ± 0.6	1.1 ± 0.3
Estimated GFR—mL/min/1.73 m^2^ *	59.2 ± 23.3	62.5 ± 19.0
**Clinical features of heart failure**		
Left ventricular ejection fraction—% *	60.3 ± 7.9	57.6 ± 7.9
Median NT-proBNP (interquartile range)—pg/mL	1064 (438–2002)	910 (464–1611)
**NYHA functional class—no. (%)**		
I	11 (2.6)	137 (2.9)
II	155 (36.3)	3706 (77.2)
III	238 (55.7)	932 (19.4)
IV	23 (5.4)	19 (0.4)
Missing data	0 (0.0)	2 (0.1)
**Medical history—no. (%)**		
Hypertension	401 (93.9)	4584 (95.6)
Diabetes	146 (34.2)	2062 (43.0)
Atrial fibrillation or flutter †	248 (58.1)	1552 (32.4)
Hospitalization for heart failure *	91 (21.3)	2306 (48.1)
**Treatment—no. (%)**		
Diuretic agent *	361 (84.5)	4585 (95.6)
ACE inhibitor or ARB	297 (69.6)	4139 (86.3)
Mineralocorticoid receptor antagonist	179 (41.9)	1239 (25.8)
Beta-blocker	314 (73.5)	3821 (79.7)

Values are given as mean ± standard deviation, or median and interquartile range, or total numbers and percent. GFR indicates glomerular filtration rate; NYHA, New York Heart Association; NT-proBNP, N-terminal prohormone of brain natriuretic peptide; ACE, angiotensin converting enzyme; ARB, angiotensin receptor blocker. * Between-cohort differences due to different inclusion and exclusion criteria (Table 1). † Number of patients with atrial fibrillation was limited to approximately 33% in PARAGON-HF.

**Table 3 jcm-09-03669-t003:** Baseline characteristics of the real-world HFpEF cohort according to eligibility for PARAGON-HF.

Characteristic	Missed Inclusion Criteria (±Fulfilled Excl. Criteria) (*Cohort 2*) n = 175	Real-World PARGON-HF (*Cohort 1*) n = 170	Fulfilled Inclusion and Met Exclusion Criteria (*Cohort 3*) n = 82	*p*-Value	Cohort 2–Cohort 1 *p*-Value	Cohort 1–Cohort 3 *p*-Value	Cohort 2–Cohort 3 *p*-Value
**Clinical parameters**							
Age—years	70.2 ± 9.4	73.3 ± 7.4	72.8 ± 7.1	**0.001**	**0.001**	0.586	**0.030**
Female sex—no. (%)	128 (73.1)	116 (68.2)	55 (67.1)	0.494	0.317	0.853	0.317
Systolic blood pressure—mmHg	140.6 ± 19.3	143.0 ± 20.6	132.2 ± 24.6	**0.001**	0.265	**0.001**	**0.008**
Heart rate—beats/min	70.7 ± 14.1	71.9 ± 14.0	72.7 ± 14.3	0.552	0.453	0.672	0.306
Body-mass index—kg/m^2^	29.9 ± 5.8	29.2 ± 5.3	32.0 ± 8.5	**0.004**	0.254	**0.007**	**0.043**
**NYHA functional class—no. (%)**				**<0.001**	**<0.001**	0.065	**<0.001**
I	11 (6.3)	0 (0.0)	0 (0.0)				
II	86 (49.1)	54 (31.8)	15 (18.3)				
III	74 (42.3)	105 (61.7)	59 (71.9)				
IV	4 (2.3)	11 (6.5)	8 (9.8)				
**Exercise capacity**							
6-min walk test (IQR)—m	385 (300–450)	323 (240–383)	252 (165–387)	**<0.001**	**<0.001**	0.070	**<0.001**
**Laboratory parameters**							
Median NT-proBNP (IQR)—pg/mL	582 (264–995)	1376 (802–2146)	2064 (1150–3533)	**<0.001**	**<0.001**	**0.001**	**<0.001**
Hemoglobin—g/dL	13.0 ± 1.6	12.4 ± 1.5	11.3 ± 2.0	**<0.001**	**<0.001**	**<0.001**	**<0.001**
Serum creatinine—mg/dL	1.1 ± 0.8	1.2 ± 0.3	1.5 ± 0.7	**<0.001**	0.291	**<0.001**	**<0.001**
Estimated GFR—mL/min/1.73 m^2^	68.5 ± 25.5	56.2 ± 16.5	47.4 ± 22.3	**<0.001**	**<0.001**	**0.002**	**<0.001**
**Medical history—no. (%)**							
Hospitalization for HF (before baseline)	8 (4.6)	49 (28.8)	34 (41.5)	**<0.001**	**<0.001**	**0.045**	**<0.001**
Arterial hypertension	163 (93.1)	159 (93.5)	79 (96.3)	0.637	0.955	0.361	0.382
Atrial fibrillation	90 (51.4)	103 (60.6)	55 (67.1)	**0.048**	0.098	0.319	**0.021**
Non-significant coronary artery disease	36 (20.6)	50 (29.4)	25 (30.5)	0.120	0.066	0.884	0.095
Diabetes mellitus type II	48 (27.4)	53 (31.2)	45 (54.9)	**<0.001**	0.465	**<0.001**	**<0.001**
Chronic obstructive pulmonary disease	37 (21.1)	46 (27.1)	33 (40.2)	**0.006**	0.188	**0.037**	**0.001**
**Concomitant medications—no. (%)**							
ACE inhibitors	46 (26.3)	59 (34.7)	22 (26.8)	0.198	0.096	0.210	0.947
Angiotensin receptor blockers	72 (41.1)	64 (37.6)	34 (41.5)	0.740	0.479	0.560	0.990
Beta blockers	121 (69.1)	131 (77.1)	62 (75.6)	0.260	0.115	0.799	0.316
Calcium channel blockers	56 (32.0)	46 (27.1)	21 (25.6)	0.443	0.298	0.807	0.285
Loop diuretics	50 (28.6)	122 (71.8)	69 (84.1)	**<0.001**	**<0.001**	**0.032**	**<0.001**
Thiazide diuretics	36 (20.6)	63 (37.1)	21 (25.6)	**0.003**	**0.001**	0.071	0.377
Mineralocorticoid receptor antagonists	37 (21.1)	87 (51.2)	54 (65.9)	**<0.001**	**<0.001**	**0.028**	**<0.001**
**Echocardiographic parameters**							
Left ventricular end-diastolic diameter—mm	43.9 ± 5.2	43.5 ± 5.4	43.8 ± 6.3	0.809	0.524	0.670	0.962
Left ventricular ejection fraction—%	60.9 ± 7.9	60.0 ± 7.8	59.5 ± 8.0	0.352	0.241	0.760	0.221
Left ventricular global longitudinal strain—%	−17.1 ± 3.8	−16.4 ± 3.8	−15.7 ± 3.9	0.094	0.193	0.312	**0.037**
Left ventricular mass index—g/m^2^	99.4 ± 23.7	100.0 ± 27.1	100.0 ± 29.3	0.977	0.834	0.986	0.886
Interventricular septum—mm	12.7 ± 2.4	12.6 ± 2.5	12.1 ± 2.2	0.235	0.665	0.181	0.083
Left atrial volume index—mL/m^2^	38.2 ± 14.7	42.5 ± 16.8	45.3 ± 17.8	**0.004**	**0.015**	0.181	**0.001**
E/A ratio	1.4 ± 0.9	1.7 ± 1.0	1.7 ± 0.9	**0.021**	**0.007**	0.627	0.122
E/e‘ ratio	13.4 ± 4.7	14.1 ± 5.7	16.4 ± 7.9	0.079	0.415	0.121	0.082
Right ventricular end-diastolic diameter—mm	34.4 ± 6.6	37.6 ± 7.3	39.7 ± 8.5	**<0.001**	**<0.001**	**0.047**	**<0.001**
Right atrial volume index—mL/m^2^	31.0 ± 16.9	41.3 ± 28.3	39.4 ± 19.6	**<0.001**	**<0.001**	0.741	**0.001**
TAPSE—mm	19.1 ± 4.7	17.7 ± 4.1	16.6 ± 4.9	**0.001**	**0.007**	0.107	**0.001**
Right ventricular tissue doppler imaging—m/s	0.13 ± 0.04	0.12 ± 0.03	0.10 ± 0.03	**<0.001**	**0.028**	**0.005**	**<0.001**
Systolic pulmonary arterial pressure—mmHg	50.6 ± 15.9	55.6 ± 16.9	65.1 ± 22.6	**<0.001**	**0.010**	**0.003**	**<0.001**
Tricuspid regurgitation velocity—m/s	3.1 ± 0.5	3.3 ± 0.6	3.6 ± 0.7	**<0.001**	**0.011**	**0.010**	**<0.001**
**Invasive hemodynamic parameters**							
Systolic pulmonary artery pressure—mmHg	48.6 ± 17.1	52.0 ± 16.8	60.6 ± 21.6	**<0.001**	0.099	**0.003**	**<0.001**
Diastolic pulmonary artery pressure—mmHg	20.1 ± 7.3	21.0 ± 6.8	25.4 ± 9.1	**<0.001**	0.312	**0.001**	**<0.001**
Mean pulmonary artery pressure—mmHg	31.2 ± 10.2	32.8 ± 9.7	38.2 ± 12.4	**<0.001**	0.208	**0.001**	**<0.001**
Pulmonary artery wedge pressure—mmHg	17.5 ± 5.9	19.5 ± 6.0	22.1 ± 7.2	**<0.001**	**0.007**	**0.011**	**<0.001**
Left ventricular end diastolic pressure—mmHg	18.5 ± 5.8	19.9 ± 5.6	20.5 ± 6.7	0.071	0.066	0.536	**0.048**
Right atrial pressure—mmHg	10.4 ± 4.7	12.2 ± 5.6	14.9 ± 6.5	**<0.001**	**0.005**	**0.003**	**<0.001**
Transpulmonary pressure gradient—mmHg	13.9 ± 8.1	13.3 ± 6.8	15.6 ± 6.9	0.154	0.528	**0.039**	0.181
Diastolic pulmonary gradient—mmHg	2.7 ± 6.2	1.5 ± 5.0	3.1 ± 5.6	0.114	0.095	0.053	0.665
Pulmonary vascular resistance—dyn·s/cm^5^	220.1 ± 128.7	219.0 ± 119.9	256.6 ± 158.7	0.167	0.944	0.081	0.108
Cardiac index Thermo—L/min/m^2^	2.69 ± 0.62	2.74 ± 0.67	2.77 ± 0.78	0.773	0.575	0.827	0.515
**Outcome parameters—no. (%)**							
Composite endpoint	30 (17.1)	65 (38.2)	40 (48.8)	**<0.001**	**<0.001**	0.112	**<0.001**
Hospitalization for heart failure	26 (14.9)	60 (35.3)	36 (43.9)	**<0.001**	**<0.001**	0.187	**<0.001**
Death from cardiac causes	6 (3.4)	23 (13.5)	16 (19.5)	**<0.001**	**0.001**	0.219	**<0.001**
Death from non-cardiac causes	14 (8.0)	18 (10.6)	9 (11.0)	0.642	0.407	0.926	0.436
Death from any causes	20 (11.4)	41 (24.1)	25 (30.5)	**<0.001**	**0.002**	0.281	**<0.001**

Values are given as mean ± standard deviation, or median and interquartile range, or total numbers and percent. Bold numbers indicate statistical significance with *p*-values < 0.05. GFR indicates glomerular filtration rate; NYHA, New York Heart Association; NT-proBNP, N-terminal prohormone of brain natriuretic peptide; IQR, interquartile range; HF, heart failure; ACE, angiotensin converting enzyme; E/e’ ratio, ratio of early transmitral blood velocity to early diastolic mitral annular velocity; TAPSE, tricuspid annular plane systolic excursion.

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
