# Peer review of "What Type of Patients Did PARAGON-HF Select? Insights from a Real-World Prospective Cohort of Patients with Heart Failure and Preserved Ejection Fraction"

_jcm, 2020, doi:10.3390/jcm9113669_

Round 1

Reviewer 1 Report

It is a good study with good data collection, comparing different patient inclusion and exclusion criteria with the previous PARAGON_HF study. The study claimed some benefits through using the new modified criteria. The data need more analyses and clarification.

Figure 2, the statistical analyses mentioned the difference between groups. It is not very clear which two groups have the difference and what their p-value is. Further analyses or further clarification need to be shown about which two groups have the statistical difference and their p-value.

Table 3, the same as above. Further analyses should be done, or further clarification needs to be shown about which two groups have the statistical difference and their p-value. Otherwise it will have some confusion.

For example, the body mass index between cohort 2 and 1 is closer, while they are smaller than cohort 3. Do they also have statistical significance? Please specify which two groups have the statistical significance.

E/A ratio: the cohort 1 and 3 are nearly the same. Do they have statistically difference? The p-value in the line showed as 0.021. It is not clear what the p-value means in this line of the table.

Diastolic pulmonary artery pressure, cohort 2 and 1 are closer. They are much lower than cohort 3. The mean pulmonary artery pressure has the similar trend. Does the difference between cohort 2 and 1 also have statistical significance? Since they are very important parameters, it is critical to have it clearly analyzed and presented. It is not clear what the p-values mean in these lines of the table.

Figure 3. Please specify which two groups showed the statistical significance. Cohort 1 and cohort 3 seem to have closer event free survival after 80 months. Do they still have statistical difference?

Overall, the data from the study are valuable. But more works need to be done with statistical analyses and data presenting.

Reviewer 2 Report

In this retrospective study, the authors compared a single-center real-world HFpEF cohort of patients with the PARAGON-HF population. Author also described their clinical characteristics and long-term prognosis comparing their cohort between eligible to those not eligible according to PARAGON-HF study criteria. Authors conclude that < 40% of real-world HFpEF patients meet eligibility criteria for PARAGON-HF and, despite reasons for optimism after PARAGON-HF, a large proportion of HFpEF patients will remain without meaningful treatment options

The hypothesis is pertinent and relevant, the design is appropriate, results are clearly presented, and the paper is well-written.

There are important limitations that deserve to be mentioned.

  1. Study population. Authors did not include as echocardiography criteria for diagnosis of HFpEF in their cohort the ESC criteria: structural alterations are a left atrial volume index (LAVI) >34 mL/m2 or a left ventricular mass index (LVMI) ≥115 g/m² for males and ≥95 g/m² for females. It could exclude patients who fulfilled HFpEF criteria. Authors should include it as a limitation and discuss.
  2. Endpoints. The authors should include in Table 3 prognosis endpoints, specify them (CV death, CV admission, non-CV death and non-CV admission) and discuss it.
  3. Endpoints. The authors should include a new Figure (or supplementary Figure) for CV and non-CV endpoints and discuss it.
  4. The authors should provide more details about the proportion of patients with paroxysmal or persistent atrial fibrillation and its relationship with beta blockers treatment. The authors should provide a comparison of clinical characteristics (BMI and atrial volume) between included vs. excluded.
  5. The authors should clarify the high proportion of NYHA III (even more than NYHA II). Which criteria were taking account? Discuss this finding.

Reviewer 3 Report

In the present paper, authors investigated the proportion of real-world HFpEF patients meeting PARAGON-HF eligibility criteria analyzing data from a prospective national registry.

The topic is very actual since to date there are no treatments with proved effectiveness in improving survival of patients with HFpEF. 

However, several concerns should be further addressed/clarified to improve the quality and the clinical relevance of the paper:

  • Authors tested the applicability of PARAGON HF eligibility criteria to a real-world cohort of patients with EF above 50%. However, PARAGON HF failed to demonstrate a beneficial effect of Sacubitril Valsartan in all HFpEF patients. A potential positive effect of Sacubitril/ Valsartan was demonstrated only in patients in the lower range of ejection fraction (45-57%). Authors should explain why they decided to test PARAGON HF eligibility criteria to all HFpEF patients rather than focusing on those with HFmrEF, which are more likely to benefit from the use of Sacubitril/Valsartan.

  • Results and limitation of PARAGON HF trial should be briefly presented in the introduction/ discussion of the paper.

  • The real-world cohort of this study consists of only 427 which is quite small. Furthermore, the number of patients from this cohort with HFmrEF should be given.

  • There are differences related to the follow-up between the PARAGON-HF trial and this study. Comparability maybe will become better if authors make it more similar.

  • In line 302, it is commented that interestingly there weren’t differences with respect to geometry and function of the LV between the three cohorts. Authors should comment or give a possible explanation of this fact.

  • Except for the ejection fraction, the entry criteria for PARADIGM-HF and PARAGON-HF were nearly identical. A recently published study assessing the proportion of ESC-EORP-HFA Heart Failure registry eligible for Sacubitril/Valsartan according to PARADIGM inclusion/exclusion criteria showed that the most difficult criterion to be met was the use of ACEi/ARB at a daily dose equivalent to >= 20 mg/Enalapril which is not required in PARAGON-HF. Has this aspect been investigate?

  • When authors say, in line 325, that patients with HFpEF have higher EF as compared to the PARAGON-HF cohort and therefore maybe be less responsive to therapy there is a contradiction because that trial already failed to show a significant beneficial effect of the treatment.

  • English should be revised.

Reviewer 4 Report

General comments:

Rettl R, et al. performed a prospective study to clarify clinical characteristics of real-world PARAGON-HF-like patients. They compared the clinical characteristics and outcome of PARAGON-HF-like patients and those who would not meet PARAGON-HF criteria. Only 170 (39.8%) patients were eligible for PARAGON-HF criteria. They showed worse outcomes compared with those who would not meet the criteria. There was a female predominance in the real-world cohort as compared to the PARAGON-HF patients, favoring the notion that more patients in the real world than in PARAGON-HF could benefit from treatment with sacubitril/valsartan.

Here is my concern that the authors should attend to:

Major points:

L81: Please describe flowchart of the selection of 427 patients with HFpEF included in your study. I think Figure 1B could be converted to the flowchart. please refer to the Figure 1 in the following article: N Engl J Med 2019;381:1609-20 (Reference No. 7).

Round 2

Reviewer 1 Report

I appreciate the efforts from the authors revising the manuscript.

This is a good manuscript detailing the modified inclusion and exclusion criteria comparing with the previous completed clinical trial. Though the perfect criteria still need to be further studied in the future for the safety and benefit of patients, no doubt this manuscript will be an important reference.

The further statistical analyses comparing each two cohorts are well done. But please specify in the methods part what statistical methods were used to compare each two cohorts.

Reviewer 3 Report

In the present paper, authors investigated the proportion of real-world HFpEF patients meeting PARAGON-HF eligibility criteria analyzing data from a prospective national registry.

The topic is very actual since to date there are no treatments with proved effectiveness in improving survival of patients with HFpEF. Despite major limitations persist, the newly submitted paper has improved a lot and all our remarks have been taken into consideration and answered. However, there are still some concerns which should be addressed before publication:

  • Regarding the use of ACEi/ARB we suggest adding: “Despite the previous use of ACEi/ARB, which was the most difficult criterion to be met when apply PARADIGM to real world, was not necessary to be enrolled in PARAGON-HF, 69.9% patients of our real-worl HFpEF cohort had been treated with ACEi/ARB, but still most of them did not met the trial criterions”.

  • In the last paragraph (conclusions) we propose to change the sentence “we conclude …” (line 602-603) with “In addition to the fact that PARAGON-HF did not demonstrate a clear benefit of the use of Sacubitril/Valsartan in patients with HFpEF, the present study show that a big proportion of real world HFpEF do not meet the trial criteria and therefore would not benefit from this treatment option”.

  • Also in the last paragraph authors should add that a study focusing on the real-world applicability of PARAGON-HF in patients with mildly reduced ejection fraction (45-57%) would be advisable since these are patients which are most likely to benefit from Sacubitril/Valsartan.
